# Breast Milk Lipids and Fatty Acids in Regulating Neonatal Intestinal Development and Protecting against Intestinal Injury

**DOI:** 10.3390/nu12020534

**Published:** 2020-02-19

**Authors:** David Ramiro-Cortijo, Pratibha Singh, Yan Liu, Esli Medina-Morales, William Yakah, Steven D. Freedman, Camilia R. Martin

**Affiliations:** 1Division of Gastroenterology, Beth Israel Deaconess Medical Center, Harvard Medical School, 330 Brookline Avenue, Boston, MA 02215, USA; dramiro@bidmc.harvard.edu (D.R.-C.); psingh6@bidmc.harvard.edu (P.S.); yliu19@seas.harvard.edu (Y.L.); jemedina@bidmc.harvard.edu (E.M.-M.); sfreedma@bidmc.harvard.edu (S.D.F.); 2Department of Neonatology, Beth Israel Deaconess Medical Center, Harvard Medical School, 330 Brookline Avenue, Boston, MA 02215, USA; wyakah@bidmc.harvard.edu; 3Division of Translational Research, Beth Israel Deaconess Medical Center, Harvard Medical School, 330 Brookline Avenue, Boston, MA 02215, USA

**Keywords:** breast milk, milk fat globule, long chain polyunsaturated fatty acids, premature infants, necrotizing enterocolitis

## Abstract

Human breast milk is the optimal source of nutrition for infant growth and development. Breast milk fats and their downstream derivatives of fatty acids and fatty acid-derived terminal mediators not only provide an energy source but also are important regulators of development, immune function, and metabolism. The composition of the lipids and fatty acids determines the nutritional and physicochemical properties of human milk fat. Essential fatty acids, including long-chain polyunsaturated fatty acids (LCPUFAs) and specialized pro-resolving mediators, are critical for growth, organogenesis, and regulation of inflammation. Combined data including in vitro, in vivo, and human cohort studies support the beneficial effects of human breast milk in intestinal development and in reducing the risk of intestinal injury. Human milk has been shown to reduce the occurrence of necrotizing enterocolitis (NEC), a common gastrointestinal disease in preterm infants. Preterm infants fed human breast milk are less likely to develop NEC compared to preterm infants receiving infant formula. Intestinal development and its physiological functions are highly adaptive to changes in nutritional status influencing the susceptibility towards intestinal injury in response to pathological challenges. In this review, we focus on lipids and fatty acids present in breast milk and their impact on neonatal gut development and the risk of disease.

## 1. Introduction

Human milk is a complex matrix of bioactive proteins, lipids, enzymes, hormones, and vitamins that collectively optimize infant development [1]. Understanding the lipid nutritional composition of breast milk provides guidance for defining adequate nutrient intake in critically ill infants, given that human breast milk fat provides almost 50% of energy intake for neonates up to 6 months of age [2]. Multiple lipid classes and compounds also found in human milk have been associated with neonatal health outcomes [3], such as adequate growth, neurocognitive development and function, regulation of inflammation and infection risk, and reduced risk of later metabolic and cardiovascular disease in adulthood. Exposure to these compounds during infancy varies, however, as it is now well understood that human milk composition is highly variable between individuals with some key determinant factors being maternal health and dietary patterns [4]. In this review, we will discuss (1) lipid and fatty acid content in breast milk, and (2) how these compounds contribute to gut development and gut health.

## 2. Lipids and Fatty Acid Composition in Human Breast Milk

### 2.1. Concentrations of Breast Milk Lipids and Fatty Acids

Breast milk fat content increases with time or “maturation”. In mothers of full-term infants, colostrum fat content is 2.2 g/100 mL, increasing to 3.0 g/100 mL in transitional milk, and 3.4 g/100 mL in mature milk [5]. In contrast, mothers with preterm neonates have higher breast milk fat concentrations [6] with values of 2.6, 3.6, and 3.9 g/100 mL in colostrum, transitional, and mature milk, respectively [7]. Saturated fatty acid content in breast milk lipids represents 53.2% in colostrum, 62.1% in transitional milk, and 58.0% in mature milk [8,9]. Table 1 shows the relative contribution of principal saturated and monounsaturated fatty acids in breast milk. Palmitic acid (C16:0), a major saturated fatty acid, provides approximately 25% of all milk fatty acids [9]. The proportion of monounsaturated fatty acids is more stable than saturated fatty acids and make up about 45%–50% of breast milk during lactation [9]. Thirty-six percent of the monounsaturated fatty acids in breast milk is oleic acid (C18:1n-9) and provides an important function in reducing the melting point of triglycerides, thus providing the liquidity required for the formation, transport, and metabolism of the milk fat globule [10,11]. Long chain polyunsaturated fatty acids (LCPUFAs) represent about 15% of the total lipids in breast milk and have been extensively studied for their developmental, cardioprotective, anti-cancer, anti-inflammatory, and antioxidant biological functions [12,13].

### 2.2. Breast Milk Fat Globules

Lipids in breast milk are present in the form of milk fat globules homogenously distributed within the aqueous phase of milk. The average size of milk fat globules, depending on the stage of lactation, varies between 0.1 and 15 μm. Breast milk fat globules are larger during the 24 h postpartum and the size reduces to similar sizes in transitional and mature milk [14,15]. Fat globules possess a core-shell structure, as illustrated in Figure 1. The membranes of a fat globule are composed of a unique tri-layer structure. The formation of a fat globule follows a coordinated sequence of synthesis and secretion. Briefly, the triglyceride core is first synthesized at the endoplasmic reticulum, and the first closely packed single layer is formed during secretion into the cytoplasm. The second, outer phospholipid bilayer is formed during secretion from the epithelial cell of the lactocyte [16,17]. These layers, also known as milk fat globule membranes (MFGM), are 8–10 nm thick and contains 70% protein, 25% phospholipid, and 5% cerebrosides/cholesterol. The phospholipid composition is distinctive between the layers, the major phospholipid components are phosphatidylethanolamine (30%) in the single layer, and phosphatidylcholine (35%) and sphingomyelin (25%) in the double layer. The MFGM also contains important proteins such as mucin-1, butyrophilin, xanthine oxidoreductase, glycoprotein bovine lactadherin 6/7, selectively placed in the double layer, which are important in infant health and have been discussed comprehensively in another review [18,19,20].

Bovine milk based infant formulas have attempted to closely mimic the lipid composition of naturally originated breast milk. However, large differences still exist in physiochemical properties between the fat globule in breast milk and the fat globule found in formula. Fat globules in conventional infant formulas are smaller (0.1–1.0 μm) in milk protein (caseins and whey proteins), dominated membranes induced by the manufacturing process of centrifugation, homogenization, and heat treatment [21,22,23]. Adding bovine milk phospholipids to an amount of 1.5% of total fat with modified processing procedures have shown to yield infant formulas with larger fat globules [23,24]. In addition, in human milk sphingomyelin is the dominant phospholipid versus phosphatidylcholine and phosphatidylethanolamine in formula [18,23]. While researchers are still making efforts to understand the functionalities of MFGM, there are already commercialized products supplementing MFGMs from bovine milk cream. The effectiveness of supplementation of MFGM in formula has been demonstrated in several independent cellular and animal studies, and clinical trials [25,26,27,28]. In mouse and rat studies, bovine MFGMs showed cognitive and neuronal development improvement and this observation was further illustrated in multiple clinical trials, suggesting bovine MFGMs have positive effects on cognitive development and protection against bacterial infection.

The downstream physiochemical properties of milk fat globules are dependent on the efficiency of fat digestion in infants. The interfacial structure, which is represented by the distribution of complex lipids in the MFGM, and the available fat area, defined as the globule size, are of primary importance for adequate lipolysis and digestion in the neonate [21,29]. In preterm infants, human milk fat globules were digested faster than preterm formulas in a digestive time range of 10 to 50 min [30]. It has been postulated that the difference in the digestion rate is caused by the variation of fat globule size and membrane composition in human milk versus preterm formulas [22]. Research in the MFGM field continues to garner interest as data indicate that breast milk fat globules could provide the right vehicle for bioactive factors, such as vitamin E and digestive enzymes, to be effectively delivered optimizing bioavailability. The MFGM fraction may have important biological roles for the development of physiological neonatal systems and immune function [9]. 

### 2.3. Complex Lipids in Breast Milk

Although about 98%–99% of lipids found in breast milk are in the form of triglycerides, other complex lipid types such as glycerophospholipids (e.g., phosphatidylethanolamine, phosphatidylcholine) and sphingolipids account for 0.2%–1.0%, or 100–400 mg/l of breast milk lipids [31]. These lipids are mostly located in MFGMs and extracellular vesicles such as exosomes in breast milk. Sphingolipids share a similar sphingosine backbone, but the type of headgroup attached determines the type of sphingolipid (sphingomyelin, glucosyl and lactosylceramides, or gangliosides). Quantification of complex lipids in human milk shows that sphingomyelin is the most abundant, making up about 36% of complex lipids, followed by glycerophospholipids, phosphatidylethanolamine (29%), and phosphatidylcholine (25%) [23,32,33]. Exosomes are also enriched with sphingolipids, particularly ceramide and sphingomyelin, and glycerophospholipids such as phosphatidylserine [34]. Like fatty acid levels, the composition of sphingolipids in breast milk can also be modified by diet. Lopez et al. demonstrated an increase in phospholipid and sphingolipid content in MFGM from cows fed with diets rich in LCPUFAs [35].

## 3. Breast Milk Lipids Enhance Neonatal Intestinal Development and Protect against Injury

The impact of lipids and fatty acids on gut development is not as well studied as in other organ systems. To date, the contribution of lipids and essential fatty acids on early postnatal gut development and subsequent host responses after an inciting event remain unknown. Understanding the changes in intestinal development in response to early priming with varying lipids and fatty acids during a pre-injurious state will be helpful to investigate the underlying mechanisms by which fatty acids may modulate the risk of intestinal injury and inflammation.

### 3.1. Saturated and Monounsaturated Fatty Acids

*Role in intestinal development and injury*. Dietary saturated and monounsaturated fatty acids have been shown to influence the microbiota diversity of human breast milk and neonatal gut [36]. Breast milk fatty acid composition changes rapidly in response to fat intake in the maternal diet. Higher levels of monounsaturated fatty acids in breast milk resulted in a decrease of *Staphylococcus, Pseudomonas*, *Lactobacillus*, and *Bifidobacterium*, while the level of saturated fatty acids in breast milk was negatively correlated with *Streptococcus*. The fatty acid composition of human milk along with specific microbial constitution likely affect the developmental programming of immune ontogeny in infants. *Bifidobacterium* and *Lactobacillus* spp. are crucial for immunological functions and mucosal barrier homeostasis, including tolerance, mucus production, tight junction expression, and T-helper cell balance [37].

In human breast milk triglycerides, the sn-2 position is commonly occupied by palmitic acid. In contrast, in palm-oil based human infant formula, palmitic acid is mainly present at the sn-1 or sn-3 position. The difference in sn- position leads to impaired absorption of calcium and fat, which negatively influence early bone accretion [38]. When human infant formula contains palmitic acid at the sn-2 position, improved absorption of fat and calcium with absorption rates similar to those seen in breastmilk-fed infants is observed [39]. Consequently, a low palmitic acid formula has been developed to reduce the amount of sn-1, 3 palmitic acids thus enabling higher absorption of calcium and fat in neonates [40,41].

Palmitic acid at the sn-2 position of the triacylglycerol backbone imparts benefits for neonatal intestinal and immunological health outcomes [42]. Lu et al. examined the effect of diet with sn-2 palmitic acids using Muc2 deficient mice, an animal model for spontaneous enterocolitis. Compared to sn-1,3 palmitic acids of triglycerides, sn-2 palmitic acids resulted in decreased intestinal injury and inflammation by upregulation of PPAR-γ, antioxidant enzymes (superoxide dismutase, glutathione peroxidase), and induction of an immunosuppressive T-regulatory cell response [43]. These data support the role of the saturated fatty acid, palmitic acid, specifically in sn-2 configuration, as being important for intestinal mucosal homeostasis, gut microbiome, and immune response [44]. However, the lack of evidence in clinical trials to establish a cause and effect relationship between the palmitic acid at sn-2 position and neonatal health outcomes, led the European Society for Paediatric Gastroenterology, Hepatology, and Nutrition to recommend that the inclusion of high sn-2 palmitic acid cannot be considered essential in human infant formula [45,46].

Among monounsaturated fatty acids, oleic acid has been shown to possess an immunomodulatory function. However, the role of oleic acid in immune responses is still controversial. An olive oil-based diet in adult mice showed improved immune responses against bacterial infection with enhanced phagocytic activity by macrophages [47,48]. In vitro, human lymphocytes treated with oleic acid resulted in increased neutral lipid accumulation thought to protect against lipid toxicity [49]. However, these studies also demonstrated oleic acid-induced cell death and necrosis mediated by caspase-3 activation. The role of oleic acid remains largely unexplored in infants [49].

### 3.2. Polyunsaturated Fatty Acids

*Role in intestinal development.* It is well-documented that in the early postnatal period of preterm infants, whole blood docosahexaenoic acid (DHA, C22:6n-3) and arachidonic acid (ARA, C20:4n-6) deficits and linoleic acid (LA, C18:2n-6) excesses occur within the first postnatal week [50]. Although these altered fatty acid profiles have been linked to the increased risk of developing bronchopulmonary dysplasia, retinopathy of prematurity, and late onset sepsis, the impact of these altered fatty acid profiles on intestinal development has just recently been described [50,51,52].

Singh et al. took advantage of the fat-1 transgenic mouse model to examine the differences in postnatal gut development in mouse pups in control, wild-type dam fed mice versus dam fed fat-1 transgenic mice [52]. Relative to the wild-type group, fat-1 mice had a greater n-3 shift in their intestinal fatty acid levels with an increase in DHA and eicosapentaenoic acid (EPA, C20:5n-3), and a decrease in ARA in the pre-weaning period. Of clinical importance is the parallel of the wild-type fatty acid levels with systemic levels observed in the postnatal period of preterm infants and the parallel of fat-1 fatty acid levels to systemic levels observed in preterm infants supplemented with enteral or parenteral fish oil [53]. This study confirmed that fatty acid exposure and intestinal levels do impact postnatal intestinal development and that a balance between both n-3 and n-6 may be critical in this early developmental period. In the pre-weaning period, while an n-3 dominant pattern increased gene expression of cell differentiation markers (EphB2, Fzd5) and fatty acid metabolism (fatty acid binding protein 2 and 6), this n-3 dominant pattern also decreased villus height over time and reduced expression of markers that inform innate host defenses, such as a reduced number of acidic mucin filled goblet cells, reduced expression of tight-junction genes (claudin 3 and 7), and reduced gene expression of muc2, trefoil factor 3, toll like receptor 9, and cathelicidin antimicrobial peptide [52]. These data suggest that that LCPUFA changes reflective of those seen in neonatal intensive care units likely influence the trajectory of postnatal intestinal development.

In a preterm piglet model evaluating the effect of an enteral complex lipid emulsion containing LCPUFAs on early postnatal fatty acid levels, it was demonstrated that an ARA:DHA ratio >1.0 compared to a ratio <1.0 uniquely prevented the postnatal deficit of ARA while also demonstrating increased ileal villus height and muscular thickness compared to the control soybean-oil and ARA:DHA <1.0 groups [54]. In contrast, parenteral nutrition did not show an effect on intestinal morphology and function in preterm piglet models, suggesting postnatal intestinal adaptation is driven more by enterally administered fatty acids versus parenteral delivery [55]. Wang et al. showed that both triglyceride (fish oil, DHA, and EPA) and phospholipid-derived n-3 LCPUFAs (DHA and EPA) enriched diets led to improved small intestine villus to crypt depth ratio in 4-week old mice. The villus to crypt depth ratio was significantly increased in the LCPUFA supplemented group compared to the control group, and this increase was more pronounced in the phospholipid-derived n-3 LCPUFAs compared to the triglyceride-derived n-3 group. Moreover, higher enrichment of gut- microbiota was observed in response to phospholipid-derived n-3 LCPUFAs [56].

*Role in intestinal injury.* Potential benefits of LCPUFAs have been reported in inflammatory bowel disease, a disease with similar pathology as necrotizing enterocolitis (NEC) [57,58]. However, human studies investigating the role of n-3 fatty acids in the prevention, treatment, and maintenance of remission inflammatory bowel disease have shown mixed results [59,60,61,62,63,64,65].

In vitro studies have consistently demonstrated anti-inflammatory actions of LCPUFAs. ARA and DHA treatment to intestinal epithelial cells blocked platelet-activating factor-induced toll like receptor 4 (TLR4) and platelet-activating factor receptor expression in intestinal epithelial cells [66]. Similarly, LCPUFA treated human fetal and adult intestinal epithelial cells demonstrated that DHA and ARA treatment led to a decreased IL-1β-induced pro-inflammatory response [67]. The protective effect of LCPUFAs in intestinal inflammatory disease could be partly explained by their effect on gut barrier function [68]. Although in vitro studies have shown mixed results about the impact of LCPUFAs on intestinal permeability [52,69,70], LCPUFA treatment to human intestinal epithelial cells resulted in improved intestinal barrier by decreasing the impairment in intestinal permeability induced by cytokines [71,72].

In vivo studies support the role of LCPUFAs in reducing inflammation and modulating the risk for NEC. Studies evaluating the effect of n-3 fatty acids and their derivatives in adult mice and a rat model of colitis showed beneficial effects by reducing the expression of inflammatory mediators [73,74,75,76,77,78]. Furthermore, Gobbetti et al. showed the protective effect of n-6, but not n-3 LCPUFA containing diet on ischemia/perfusion induced intestinal injury in an adult murine model. Three-week-old mice supplemented with n-6 enriched diet for 9 weeks showed reduced intestinal damage and inflammation and this effect was mediated by increased level of lipoxin A4, suggesting the anti-inflammatory role of lipoxin A4 in ischemia/perfusion induced intestinal injury [79]. These results suggest that ARA is not only a precursor of pro-inflammatory lipid mediators but also plays a critical role in generating anti-inflammatory lipid mediators [80]. Altogether, the balance of n-3 and n-6 LCPUFAs plays a critical role in regulating intestinal pathophysiology and inflammation.

Several animal studies using experimental models of NEC have shown that LCPUFA supplementation results in reduced NEC incidence and its severity by regulating multiple pathways associated with intestinal inflammation/injury and necrosis, including TLR4, platelet-activating factor, and nuclear factor-κB [81,82,83]. Preterm rats born to dams who were fed a DHA or EPA enriched diet and subjected to an NEC induction protocol showed significant decreased NEC-like colitis incidence [84]. In a different study, LCPUFA supplementation significantly reduced the incidence of NEC in a neonatal rat model compared with controls by downregulating phospholipase A(2)-II and platelet-activating factor receptor at 24 and 48 h, respectively [81]. Compared to a control formula, three different LCPUFA supplementation strategies (ARA and DHA, egg phospholipid, DHA) showed reduced NEC incidence in a neonatal rat model of NEC by downregulating TLR4 expression [66]. Similarly, young mice supplemented with 10% fish oil for 4 weeks showed protective effects against hypoxia-induced NEC with a reduction in platelet-activating factor as well as leukotriene B4 [85].

Several clinical trials have evaluated the impact of LCPUFAs in term and preterm infants. However, the data about the role of LCPUFAs in NEC is limited. Furthermore, in many of these studies, the assessment of the risk of NEC with nutritional intervention was a secondary outcome analysis. Differences in LCPUFA formulation, dosing, and study populations results in inconclusive data about the use of LCPUFA in the prevention of NEC. Despite promising results of LCPUFAs in experimental NEC in animal models, results in preterm human infants are mixed and limited.

A summary of LCPUFA supplemented feedings and NEC risk in preterm infants is shown in Table 2. The systemic review by Smithers et al. did not find any benefit of n-3 LCPUFA supplemented formula in the risk of NEC. In this work, the studies were not limited to preterm infants who are at the highest risk of developing NEC [86]. Fewtrell et al. included infants who received human milk, which could be a confounding factor in this study [87]. In the Carlson et al. study, no infant received human milk and the preterm infants were randomly assigned into two groups receiving either experimental formula containing egg phospholipids (0.13% DHA and 0.41% ARA) or control group. The egg phospholipid-containing formula resulted in a significant decrease in NEC incidence compared to the control formula [88]. Currently, this is the only single-center human study demonstrated to decrease NEC incidence in response to LCPUFA supplementation. It is important to note that none of the trials looked at NEC incidence as a primary outcome. In addition, what is unclear is whether a potential benefit would be seen in a population of largely human milk fed infants, which is the current standard.

### 3.3. Milk Fat Globule Membranes

*Role in intestinal development*. MFGM proteins have been identified [18] that exert multiple biological functions critical for intestinal development and health. Mucin 1 interacts with molecule-3-grabbing non-integrin, expressed by dendritic cells, in the infant gastrointestinal tract and subsequently inhibits binding of certain intestinal pathogenic bacteria to dendritic cells [97]. Moreover, mucin 1 demonstrates anti-viral properties against rotavirus infection [98]. Xanthine oxidoreductase, a predominant protein of MFGM, possesses antimicrobial activity and has been shown to inhibit the growth of *Escherichia coli* and *Salmonella enteritides* by generating superoxide and peroxide, thus providing protection to the neonatal gut [99]. Lactadherin, also known as milk fat globule–EGF factor 8, aids in the clearance of dead cells by macrophages through phagocytosis [100]. Exogenous administration of lactadherin decreased the inflammatory response and injury by enhancing phagocytosis of apoptotic cells in experimental sepsis and ischemia/perfusion injury models [101,102,103]. Furthermore, lactadherin deficient mice showed more severe injury in response to dextran sulfate sodium, which induces colitis, and the administration of human lactadherin attenuated the dextran sulfate sodium injury [104]. Results from in vitro and in vivo studies showed that lactadherin deficiency results in delayed epithelial cell renewal and turnover [105]. Together, these results suggest the importance of lactadherin in intestinal homeostasis and imply that the provision of lactadherin in human or bovine milk could be beneficial against intestinal injury/inflammation. Butyrophilin, another quantitatively major protein in human MFGM, has been considered as a potentially important regulator for immune function. In addition to its effect on the secretion of milk fat globules [106], butyrophilin induces apoptosis and promotes T cell response [107]. Finally, in rat pups, the addition of MFGM to formula resulted in an enteric microbiome and intestinal development pattern (measured by villus height, crypt depth, crypt cell proliferation, paneth and goblet cell counts, and tight junction proteins) more similar to breastfed rats compared to rats fed with formula without MFGM [108]. Collectively, these results suggest the important role of MFGM and MFGM proteins in shaping the gastrointestinal tract immune system and maintaining immune homeostasis.

*Role in intestinal injury*. MFGM supplemented formula in rat pups protects against *C. difficile* toxin [108] and LPS-induced intestinal inflammation [109]. In the latter study, MFGM supplementation protected against histological changes in the intestine, reduced inflammatory cytokines, and increased tight junction proteins. In an asphyxia, cold stress model of NEC in neonatal rat pups, MFGM supplementation reduced NEC incidence and severity with a concurrent reduction in TLR4 expression [110]. Clinical trials of MFGM supplementation have not been conducted in preterm infants. In 6–12-month-old infants supplemented daily with MFGM, the number of bloody diarrheal episodes was reduced significantly by almost 50% [111]. Rapidly evolving and emerging evidence from animal model studies and clinical trials indicates the potential of beneficial effects of the combination of bioactive compounds or any specific component of MFGM to optimize postnatal intestinal development in the preterm infant and to protect against disease.

### 3.4. Complex Lipids

*Role in intestinal development and injury*. Breast milk provides complex lipids through the delivery of MFGMs and exosomes. Published data support the role of extracellular vesicles on intestinal development and protection from intestinal injury such as NEC [112,113,114]. When evaluated in totality versus in individual components, human milk-derived exosomes attenuate oxidative stress-induced cell death in intestinal epithelial cells [115] and enhance proliferation and migration of intestinal epithelial cells in preterm infants compared to those with full term birth [116].

Breast milk sphingolipids, such as sphingomyelin and gangliosides, are important modulators of neonatal intestinal development, the establishment of the gut microbiome, and inflammation [117]. Sphingomyelin is digested by nucleotide phosphodiesterase pyrophosphatase 7 (NPP7), a brush border enzyme of the intestinal epithelium, and generates ceramide, sphingosine, and sphingosine-1-phosphate [118,119]. In contrast to sphingomyelin and ceramide, sphingosine is rapidly absorbed and largely converted to palmitic acid in the mucosa [120]. NPP7 is specifically expressed in intestinal mucosa and is highly expressed in the middle part of the jejunum and the lower colon.

NPP7 possesses phospholipase C activity against platelet activating factor, a pro-inflammatory lipid mediator produced by gut epithelial cells [121]. Higher levels of platelet activating factor have been shown in inflammatory bowel disease, ischemic colitis and NEC [118]. Intrarectal administration of recombinant NPP7 significantly reduced the intestinal injury and inflammation against colitis in an adult rat model [122]. It is possible that NPP7 expression is dependent on gestational age or the changes in enzymatic activity that occur later after birth, which may predispose the preterm infant to intestinal injury. However, similar expression of NPP7 was observed in the meconium of term and preterm infants [123]. The anti-inflammatory benefits of sphingomyelin might be mediated through the induced changes in the microbiome. In an adult murine model, a diet containing 0.25% (wt/wt) milk sphingomyelin had lower fecal Gram-negative bacteria and higher fecal *Bifidobacterium* compared to mice fed with a high fat diet, and these changes were accompanied by reduced serum lipopolysaccharide levels in the sphingomyelin group [124].

Gangliosides have been reported to reduce pro-inflammatory signaling in the intestine [125] and protect the bowel in an infant model of necrotizing enterocolitis [126]. Weaning rats fed with ganglioside-enriched diets demonstrated increased ganglioside levels in the intestinal brush border and reduced levels of the pro-inflammatory mediator platelet activating factor [127]. In a follow-up study, rats fed with dietary gangliosides compared to controls exhibited reduced expression of pro-inflammatory mediators such prostaglandin E2, LTB4, IL-1β, TNF-α in the intestinal mucosa following lipopolysaccharide-induced inflammation [128]. Similar to sphingomyelin, some of this mediation may be due to induced microbiome changes in the gut. In preterm infants, fecal *Escherichia coli* counts were lower and *Bifidobacterial* counts were higher in infants fed with a ganglioside-supplemented diet compared to infants fed a control milk formula [129].

Together, these studies support the role of sphingolipids, particularly sphingomyelin and gangliosides, as important mediators of intestinal development and protection against intestinal injury.

## 4. Scientific Gaps and Future Directions

Dietary lipids and fatty acids are of critical importance in several developmental processes such as immune responses, organogenesis, and central nervous system development. Although after a term pregnancy adipose stores and human milk continue to supply the infant with critical lipids and fatty acids, preterm infants do not have these sustainable sources with a lack of adipose tissue and minimal enteral feeding volumes. To prevent postnatal deficits of these critical nutrients, the preterm infant will likely require additional dietary supplementation. The specific content of such supplementation and the chemical composition to ensure adequate absorption remains to be defined. Based on the available data, infant formula should provide both ARA and DHA, and in an ARA:DHA ratio >1. Providing solely DHA or DHA exceeding ARA may induce undesirable health outcomes in infants, leading to adverse effects on growth and immune development [130,131].

The precise mechanisms by which lipids and fatty acids mediate their effects on the developing intestine still need to be fully characterized. It is possible that dietary intervention may have different effects across different exposure times and different clinical contexts (during acute illness versus convalescence). As a result, studies that examine timing, composition, and dosing, including desirable target levels, are needed.

The role of human milk-derived vesicles, including the human milk fat globule and exosomes, may reveal an opportunity to present multiple critical molecules simultaneously and ensuring delivery and bioavailability to the intended site.

The biggest challenge in translational research is reconciling the disparate results obtained from animal models versus human clinical trials. Establishing animal models that better reflect the preterm neonatal experience, as well as refinement of humanoid model systems, will be essential in bridging the gap from bench to bedside.

## 5. Conclusions

The lipid and fatty acid content in human milk inform investigators and clinicians of the important nutrient pathways that facilitate growth, development, and resistance to disease. The preterm infant is uniquely vulnerable to nutrient deprivation, and parenteral and enteral feedings alone may not be sufficient to close the nutrient gaps that develop early after delivery. The science of breast milk will likely open new avenues of therapeutic options to minimize the health consequences of such nutrient gaps. An improvement in the experimental model systems will iteratively close the gap in translation from bench to bedside.

## Figures and Tables

**Figure 1 nutrients-12-00534-f001:**
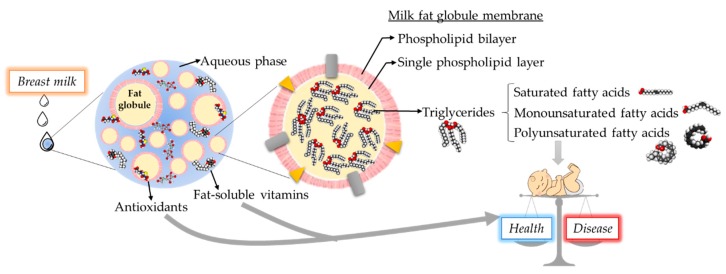
Breast milk fat components and relationship with neonatal health-disease balance. Scheme of fat globule illustrating of the core-shell structure.

**Table 1 nutrients-12-00534-t001:** Fatty acid profile of breast milk across lactation stages.

	Term Infants			Preterm Infants		
	Colostrum	Transitional	Mature	Colostrum	Transitional	Mature
*Saturated Fatty Acids*						
Caprylic acid (C8:0)	0.07–0.19	0.2–0.31	0.2–0.3	0.03–0.03	0.09–0.11	0.16–0.16
Capric acid (C10:0)	0.5–1.04	1.2–1.6	1.5–1.8	0.09–0.09	1.0–1.7	1.2–2.1
Lauric acid (C12:0)	2.8–3.5	5.4–6.6	5.7–6.5	3.2–4.6	5.7–7.5	5.7–8.1
Myristic acid (C14:0)	5.4–6.0	6.6–7.5	6.5–7.1	5.8–7.2	8.0–9.2	7.4–9.0
Palmitic acid (C16:0)	24.3–25.5	21.9–23.3	21.7–22.7	22.5–24.1	21.5–23.5	20.9–22.3
Stearic acid (C18:0)	6.2–6.6	6.1–6.7	6.3–6.6	5.8–6.5	6.0–6.9	6.2–7.1
Arachidic acid (C20:0)	0.19–0.25	0.20–0.32	0.20–0.26	0.16–0.18	0.15–0.15	0.20–0.30
*Monounsaturated Fatty Acids*						
Myristoleic acid (C14:1n-5)	0.13–0.23	0.19–0.25	0.18–0.22	0.11–0.13	0.22–0.22	0.21–0.21
Palmitoleic acid (C16:1n-7)	1.9–2.2	2.0–2.4	2.2–2.4	1.7–1.8	2.1–2.5	2.0–2.5
Oleic acid (C18:1n-9)	34.7–35.9	31.2–33.2	32.2–33.6	30.6–33.7	30.5–34.3	31.7–36.7
Vaccenic acid (C18:1n-7)	2.6–2.8	1.9–2.0	1.7–2.1	2.3–2.4	2.5–2.6	2.1–2.2
Erucic acid (C22:1n-9)	0.20–0.24	0.14–0.28	0.10–0.12	0.16–0.16	0.10–0.14	0.08–0.05
n-*3 Polyunsaturated fatty acids (*n-*3 LCPUFAs)*						
α-Linolenic acid (C18:3n-3)	0.74–0.90	0.84–1.06	0.91–1.03	0.69–1.09	0.70–1.02	0.85–1.13
Eicosapentaenoic acid (C20:5n-3)	0.08–0.12	0.11–0.17	0.08–0.10	0.06–0.10	0.10–0.16	0.08–0.16
Clupanodonic acid (C22:5n-3)	0.27–0.33	0.19–0.25	0.14–0.16	0.30–0.34	0.24–0.36	0.16–0.24
Docosahexaenoic acid (C22:6n-3)	0.47–0.55	0.40–0.52	0.28–0.34	0.43–0.71	0.47–0.67	0.31–0.49
n-*6 Polyunsaturated fatty acids (*n-*6 LCPUFAs)*						
Linoleic acid (C18:2n-6)	13.5–15.3	13.4–14.8	14.3–15.7	13.7–16.3	11.4–13.6	12.3–14.4
γ-Linolenic acid (C18:3n-6)	0.07–0.11	0.10–0.18	0.14–0.20	0.07–0.07	0.09–0.13	0.11–0.21
Eicosadienoic acid (C20:2n-6)	0.82–0.96	0.53–0.63	0.35–0.41	0.89–0.95	0.28–0.30	0.24–0.24
Dihomo-γ-Linolenic acid (C20:3n-6)	0.56–0.64	0.46–0.52	0.39–0.43	0.69–0.81	0.47–0.55	0.40–0.50
Arachidonic acid (C20:4n-6)	0.73–0.81	0.61–0.69	0.45–0.51	0.68–0.90	0.54–0.68	0.48–0.58
Docosatetraenoic acid (C22:4n-6)	0.29–0.39	0.19–0.25	0.09–0.11	0.44–0.49	0.22–0.22	0.13–0.17
Adrenic acid (C22:5n-6)	0.13–0.21	0.09–0.13	0.06–0.10	0.15–0.17	0.05–0.05	0.05–0.09

Data shows the relative proportion in total lipids (%) between mothers who had term and preterm infants. Colostrum = 0–5 days of postnatal life (DPL); Transitional = 6–15 DPL; Mature = 16–60 DPL. Data abstracted from [6].

**Table 2 nutrients-12-00534-t002:** Human studies of long-chain polyunsaturated fatty acid (LCPUFA) supplementation in preterm infants and necrotizing enterocolitis (NEC) risk.

Reference	Study Design	Population	n	Powerful and Prevalence of NEC	Principal Finding in NEC
Smithers et al. (2008) [86]	Systematic review	<37 GA	1333	RR = [0.62–2.04]	No benefit of n-3 LCPUFA supplemented formula
Zhang et al. (2014) [89]	Systemic review	<32 GA	900	RR = [0.23–1.10]	No benfit of n-3 LCPUFA supplementation
*Double-blinded randomized clinical trials*					
Carlson et al. (1998) [88]	Formula supplemented with 0.41% ARA + 0.13% DHA	<32 GA BW between 725–1375 g	119	Control = 17.6% Experimental = 2.9%	Significantly decreased
Fewtrell et al. (2002) [90]	Formula supplemented with 0.31% ARA + 0.17% DHA	<37 GA BW <1750 g	197	Control = 11% Experimental = 19%	No significant difference
Innis et al. (2002) [91]	BM supplemented with DHA BM supplemented with ARA + DHA	BW between 846–1560 g	194	Control = 1.6% Experimental = 1.5%	No significant difference
Fewtrell et al. (2004) [87]	Formula supplemented with 0.31% ARA + 0.17% DHA	<35 GA BW ≤2000 g	238	Control = 2% Experimental = 4%	No significant difference
Clandinin et al. (2005) [92]	Formula supplemented with DHA + ARA	<35 GA	361	Control = 3% Experimental = 5%	No significant difference
Henriksen et al. (2008) [93]	BM supplemented with 6.7% ARA + 6.9% DHA	BW <1500 g	141	Control = 3% Experimental = 1.5%	No significant difference
Makrides et al. (2009) [94]	High DHA (1%) Low DHA (0.3%)	<33 GA	657	Adj. OR = [0.87–5.22]	No significant difference
Collins et al. (2016) [95]	Formula supplemented with different doses of DHA	<30 GA	53	Control = 9% Experimental = 9%	No significant difference
Collins et al. (2017) [96]	BM supplemented with 60 mg/kg/day DHA	<29 WGA	1273	Adj. OR = [0.79–1.69]	No significant difference

In the double-blinded randomized clinical trials, the control group was no supplementation feeding. Breast milk (BM); birth weight (BW); weeks of gestational age (GA); the relative risk (RR) or adjusted odd ratio (OR) shown as 95% confidence interval.

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
