# Peer review of "Breast Milk Lipids and Fatty Acids in Regulating Neonatal Intestinal Development and Protecting against Intestinal Injury"

_nutrients, 2020, doi:10.3390/nu12020534_

Round 1
Reviewer 1 Report
In the Review “Breast milk lipids and fatty acids in regulating neonatal intestinal development and protecting against intestinal injury,” the authors have reviewed the literature available on the composition of human breast milk lipids and the function of different fatty acids and milk fat globule membrane proteins in neonatal intestinal development. They further discuss how the different human breast milk components protect the neonatal intestine from injuries. The authors conclude the review with different questions that need to be answered regarding infant formula supplementation, the precise mechanisms by which lipids and fatty acids mediate their effects on the developing intestine and the biggest challenge that must be addressed in translational research to reconcile the disparate results obtained from animal models versus human clinical trials. The review is well written and can be accepted for publication. However, I would like the authors to address a few minor corrections in the review.
Page 4 Line 146, “In human breast milk, the sn-2 position is commonly occupied by palmitic acid”. It would be more appropriate to write “In human breast milk lipids, the sn-2 position is commonly occupied by palmitic acid” Page2, Line 52-54, breast milk fat content are mentioned as 2.2/100ml and so on. Please mention the units in which the milk lipids are represented per 100ml
Author Response
Thank you for reviewing our manuscript and your suggestions. Kindly find our responses to your suggestions in the attached PDF.
Reviewer 2 Report
This paper nicely discusses breast milk lipids and fatty acids in regulating neonatal intestinal development and protecting against intestinal injury, and hereby covers the aim of the journal and the subject investigated is of worldwide interest.
The text is well organized and the different issues properly discussed, although some latest literature is missing. I suggest that the few additional references will benefit the manuscript. I hope that the authors will incorporate these into the manuscript.
Koletzko B, Bergmann K, Brenna JT, Calder PC, Campoy C, Clandinin MT, Colombo J, Daly M, Decsi T, Demmelmair H, Domellöf M, FidlerMis N, Gonzalez-Casanova I, van Goudoever JB, Hadjipanayis A, Hernell O, Lapillonne A, Mader S, Martin CR, Matthäus V, Ramakrishan U, Smuts CM, Strain SJJ, Tanjung C, Tounian P, Carlson SE. Should formula for infants provide arachidonic acidalong with DHA? A position paper of the European Academy of Paediatrics and the Child Health Foundation. Am J Clin Nutr. 2020 Jan 1;111(1):10-16. Norris GH, Milard M, Michalski MC, Blesso CN. Protective properties of milksphingomyelin against dysfunctional lipid metabolism, gut dysbiosis, and inflammation. J Nutr Biochem. 2019 Nov;73:108224. doi: 10.1016/j.jnutbio.2019.108224. Epub 2019 Aug 15. Miliku K, Duan QL, Moraes TJ, Becker AB, Mandhane PJ, Turvey SE, Lefebvre DL, Sears MR, Subbarao P, Field CJ, Azad MB. Human milkfatty acid composition is associated with dietary, genetic, sociodemographic, and environmental factors in the CHILD Cohort Study. Am J Clin Nutr. 2019 Dec 1;110(6):1370-1383. doi: 10.1093/ajcn/nqz229. Wei W, Yang J, Yang D1, Wang X1, Yang Z2, Jin Q, Wang M1, Lai J2, Wang X. Phospholipid Composition and Fat Globule Structure I: Comparison of Human Milk Fat from Different Gestational Ages, Lactation Stages, and Infant Formulas. J Agric Food Chem.2019 Dec 18;67(50):13922-13928. doi: 10.1021/acs.jafc.9b04247. Epub 2019 Dec 10. Jiang T 1, Liu B , Li J , Dong X , Lin M , Zhang M , Zhao J , Dai Y , Chen L . Association between sn-2 fatty acid profiles of breast milk and development of the infant intestinal microbiome. Food Funct.2018 Feb 21;9(2):1028-1037. doi: 10.1039/c7fo00088j. Vegge A, Thymann T, Lauritzen L, Bering SB, Wiinberg B, Sangild PT. Parenteral lipids and partial enteral nutrition affect hepatic lipid composition but have limited short term effects on formula-induced necrotizing enterocolitis in preterm piglets. Clin Nutr. 2015 Apr;34(2):219-28. doi: 10.1016/j.clnu.2014.03.004. Epub 2014 Mar 19.
Minor points
Line 52 Revise „colostrum breast milk” to „colostrum”
Line 87 The Figure 1 applies to breast milk and the presence of a milk bottle in the scheme can be misleading.
Line 146 Revise „In human breast milk” to „In human breast milk triglycerides”
Author Response

(The authors gave the same response as above.)

Reviewer 3 Report
Dear authors,
your review entitled "Breast milk lipids and fatty acids in regulating neonatal intestinal developement and protection against intestinal injury" is comprehensive and nicely written. The article looks scientifically sound and backed-up by an impressive number of references.
Therefore I only have a few minor corrections to mention:
-Lines52-54: unit missing for the colostrum breast milk fat content (x.x g/100ml?)
-Line 202: EPA acronym has not been defined in plain text.
-Line 359: there is one "milk" too much in " human milk fat milk globule", I guess
Author Response

(The authors gave the same response as above.)

Reviewer 4 Report
The review article is very well written and interesting. It includes tables and figures useful for the readers.
I do not have ant specific point to improve the manuscript that will be highly cited and useful for future researches.
As a minor point, I would suggest to carefully check the article for minor typos.
Author Response
Thank you for your time to review this manuscript. We carefully checked the article for minor typos and corrected them.